# A Topology Preserving Gridding Method for Vector Features in Discrete Global Grid Systems

**Liangchen Zhou, Wenjie Lian, Yudi Zhang and Bingxian Lin ***

Key Laboratory of Virtual Geographic Environment, Nanjing Normal University, Nanjing 210023, China;
zhoulch@njnu.edu.cn (L.Z.); 161302072@stu.njnu.edu.cn (W.L.); 181302111@stu.njnu.edu.cn (Y.Z.)
*  Correspondence: 09345@njnu.edu.cn; Tel.: +86-13951652903

**Abstract:** Topological distortion seriously affects spatial cognition. To solve this problem caused by the integration of vector features in discrete global grid systems (DGGs), a topology-preserving gridding method for vector features is proposed. The method proposed determines the topological distortion according to the relationship between grid cells and then increases the local resolution of vector features by employing the multi-level resolution characteristic of DGGs, to repair three kinds of topological distortions. Experimental results show that the proposed method can effectively maintain the topological relationship between the original vector features, and the amount of data is stable, thus ensuring the correct integration of vector features in the DGGs.

**Keywords:** discrete global grid systems; vector feature; gridding; data integration; topological relationship

---

## 1. Introduction

A discrete global grid system (DGG) divides the earth's surface into seamless and nonoverlapping multi-level regional unit sets, which are used to fit the earth's surface [1–3] at different resolutions and can standardize the integration and analysis of massive spatial data [4] at any resolution. A DGGs can be thought of as a framework of Digital Earth [5] for the integration of spatial data [6,7] and earth system modeling [8]. In order to realize the grid representation of geographical objects, integration of vector features, that is, the discretization of vector features into grid cells of corresponding scales according to certain criteria, must be performed and is the core problem of discrete global grid systems research [9].

Because of the difference of the data model itself, the gridding of vector features is inherently a lossy process, which leads to distortion of geometric properties and topological relations [10]. In the literature, scholars usually focus on the preservation of geometric properties such as angle, length, and area in vector feature integration [11] but ignore the preservation of topological relations.

The topological relationship represents the spatial relationship between geographical entities; thus, the wrong topological relationship will seriously affect spatial cognition. In Figure 1, the bay becomes an inner lake isolated from the ocean after gridding, and this is a serious topological distortion. In the simulation of ocean model, if this topology distortion occurs in the channel, it will lead to a huge difference between the simulation results and the actual situation.

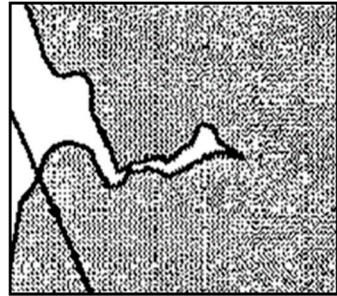 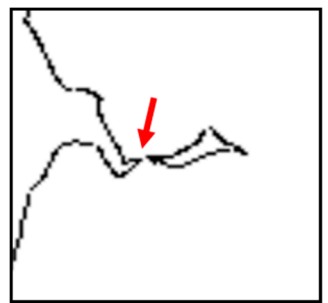

(a) The actual boundary of the original vector feature

(b) The boundary after gridding

**Figure 1.** Topological distortion in a vector feature grid [1].

In this paper, a topology-preserving gridding method for vector features in DGGs is proposed. The following sections are arranged as follows: The second section introduces the basic idea of this method, the third section defines different types of topological distortion, the fourth and fifth sections describe the concrete implementation steps, and experimental verification of the proposed method, and the sixth, and seventh sections discuss and summarize this paper.

## 2. Related Work

The discretization of vector features into grid cells is the core problem of discrete global grid systems research. In vector features, the gridding of point features is relatively simple and can be expressed by grid cells corresponding to their scales [12], while for line features and polygon features all the grid cells covered according to the scale must be determined. In order to solve this problem, scholars have ported gridding algorithms in a planar grid to a spherical space [13,14]. Some scholars noted the gridding of vector features is a lossy process, proposed geometric-preserving method of angle, length, and area [11] but ignore the preservation of topological relations.

In Traditional GIS, scholars have observed a change of topological relationships after vector feature gridding [15,16] and proposed a corresponding amendment method [17]. However, this correction method directly modifies the ownership of grid cells with topological distortion, and although it corrects the topological distortion after rasterization, it introduces new geometric deformations. Although this problem can be solved by increasing the grid resolution, in a flat grid system, only a single resolution of the grid cell is present; thus, the local grid resolution cannot be increased, and increase of the global grid resolution results in an exponential increase in data volume.

## 3. Basic Idea

Discrete global grids (DGGs) are a typical multi-resolution grid system. If vector features with changed topological relationships can be identified, the local resolution can be improved by employing the multi-resolution characteristic of the DGGs, and topological distortion can be repaired without introducing new geometrical deformation.

In this paper, based on the 9-intersection model [18], the topological distortions that may occur in the gridding of vector features in DGGs are classified, and the corresponding topology-preserving gridding methods for vector features are designed for different types of distortions. The 9-intersection model formally captures topological relations between two spatial objects through the geometric intersections of the objects' interiors, exteriors, and boundaries. For all kinds of vector features, this method can be divided into three main steps: initial gridding, topological distortion detection, and topological distortion repair, as shown in Figure 2.

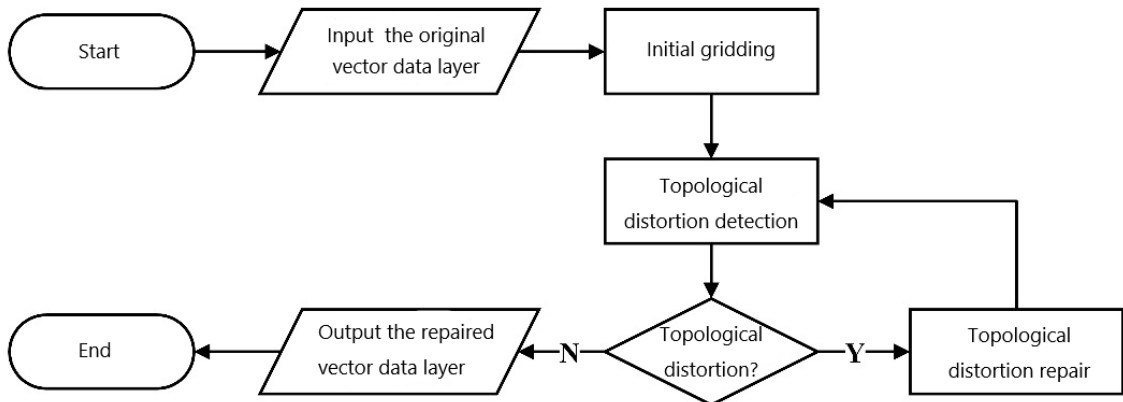

**Figure 2.** The overall process of this method.

In the initial gridding step, the topological relationship among the features in the original vector feature is first recorded, and then, the vector features are discretized into DGGs through different strategies according to the scale of the original vector feature. In the topology distortion detection step, according to the topological relationship among the features after the initial gridding, the features of any topological relationship change are determined, and the corresponding grid cells generating topological distortion are recorded. In the topological distortion repair step, the grid cells with topological distortion are locally divided, and their level is improved according to the multi-resolution characteristic of DGGs. Topological distortion detection and repair will perform repeatedly until all topology distortion is repaired.

## 4. Topological Distortion Classification

In practical application, vector features are usually organized according to the type of feature. Therefore, this section studies the topological distortion according to the type of feature.

### 4.1. Topological Distortion of Point Features

According to the 9-intersection model, a point feature is a zero-dimensional object. There are two kinds of topological relationships between point features, equals as shown in Equation (1) and disjoint as shown in Equation (2).

$$R_9(Point,\ Point) = \begin{bmatrix} T & * & * \\ * & * & * \\ * & * & * \end{bmatrix} \tag{1}$$

$$R_9(Point,\ Point) = \begin{bmatrix} F & * & * \\ * & * & * \\ * & * & * \end{bmatrix} \tag{2}$$

If the spatial coordinates of the two original point features are exactly the same, then the two point features are topologically equal; otherwise, they are disjoint from each other. After the gridding of the point feature, the two original disjoint point features may be transformed into the same grid cell, and the topological relationship of the point feature object changes from disjoint to equal, as shown in Figure 3.

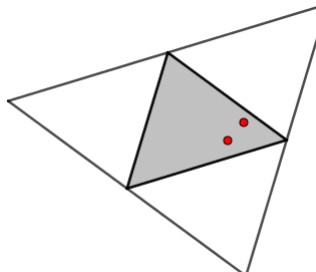

**Figure 3.** Change from disjoint to equal after gridding of point features.

*4.2. Topological Distortion of Line Features*

According to the 9-intersection model, the topological relationship between line features is disjoint as shown in Equation (3) or overlaps as shown in Equation (4).

$$R_9(\textit{Polyline, Polyline}) = \begin{bmatrix} F & F & * \\ F & F & * \\ * & * & * \end{bmatrix} \tag{3}$$

$$R_9(\textit{Polyline, Polyline}) \neq \begin{bmatrix} F & F & * \\ F & F & * \\ * & * & * \end{bmatrix} \tag{4}$$

After the gridding of line features, the two line features may contain common grid cells, and the topological relationship of the line features changes from disjoint to overlaps, as shown in Figure 4.

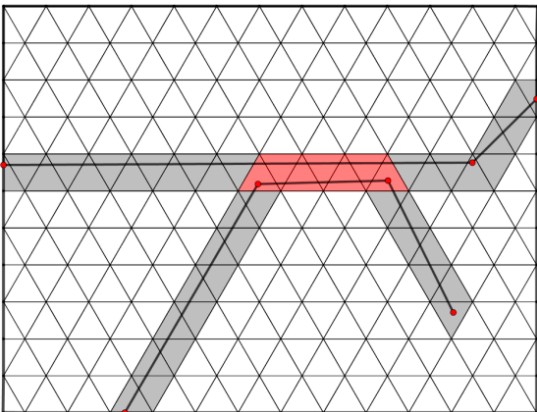

**Figure 4.** Change from disjoint to overlaps after gridding of line features.

*4.3. Topological Distortion of Polygon Features*

The topological relationship between polygon features is more complex, but in practical application, there are usually no overlaps between the various features, such as land cover data. At this time, according to the 9-intersection model, there are only two kinds of topological relations between the polygon features, meets as shown in Equation (5) and disjoint as shown in Equation (6) [17].

$$R_9(\textit{Polygon, Polygon}) = \begin{bmatrix} F & * & * \\ * & T & * \\ * & * & * \end{bmatrix} \tag{5}$$

$$R_9(Polygon,\ Polygon) = \begin{bmatrix} F & F & * \\ F & F & * \\ * & * & * \end{bmatrix} \tag{6}$$

The gridding of a single polygon feature may occur when the polygon feature disappears or is divided into multiple polygon features [17]. Detailed descriptions are provided as follows:

### 4.3.1. Meets to Disjoint

If the two original features (i.e., Polygon A and C of Figure 5a) are connected, and polygons associated with one of the features (i.e., Polygon C of Figure 5a) is in a narrow area, where is connected to a third polygon feature (i.e., Polygon B of Figure 5b), the topological relationship between the two polygons may change from meet to disjoint, as shown in Figure 5.

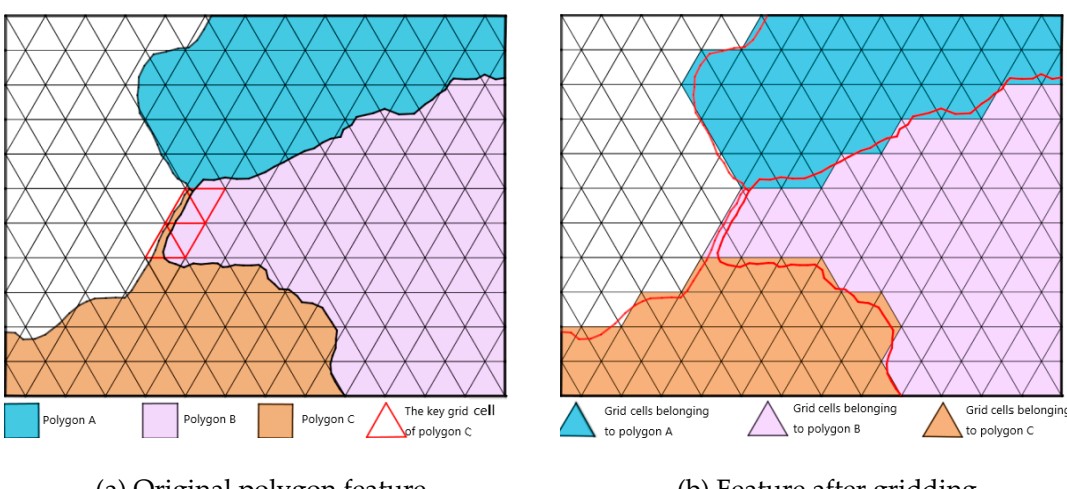

(a) Original polygon feature                    (b) Feature after gridding

**Figure 5.** Change from meet to disjoint after gridding of polygon features.

### 4.3.2. Disjoint to Meets

If the two original polygon features (i.e., Polygon A and B of Figure 6a) are disjoint, and the two features are meet, another polygon feature with a narrow area (i.e., Polygon C of Figure 6a). After gridding, the topology relationship of the two polygon features (i.e., Polygon A and B of Figure 6b) may change from disjoint to meets, as shown in Figure 6b.

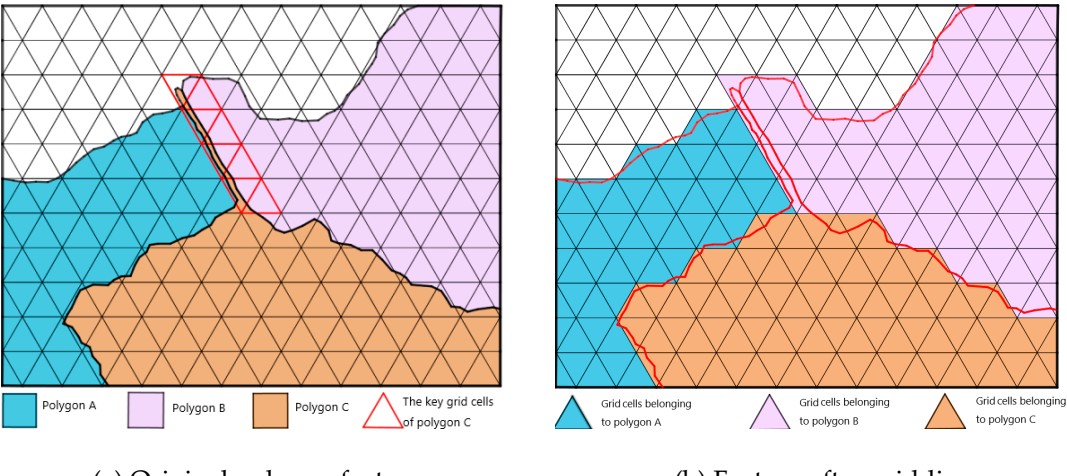

(a) Original polygon feature                    (b) Feature after gridding

**Figure 6.** Change from disjoint to meet after the gridding of polygon features.

### 4.3.3. Polygon Disappearance

Relatively small polygon features (i.e., Polygon B of Figure 7a) may disappear directly after gridding, as shown in Figure 7.

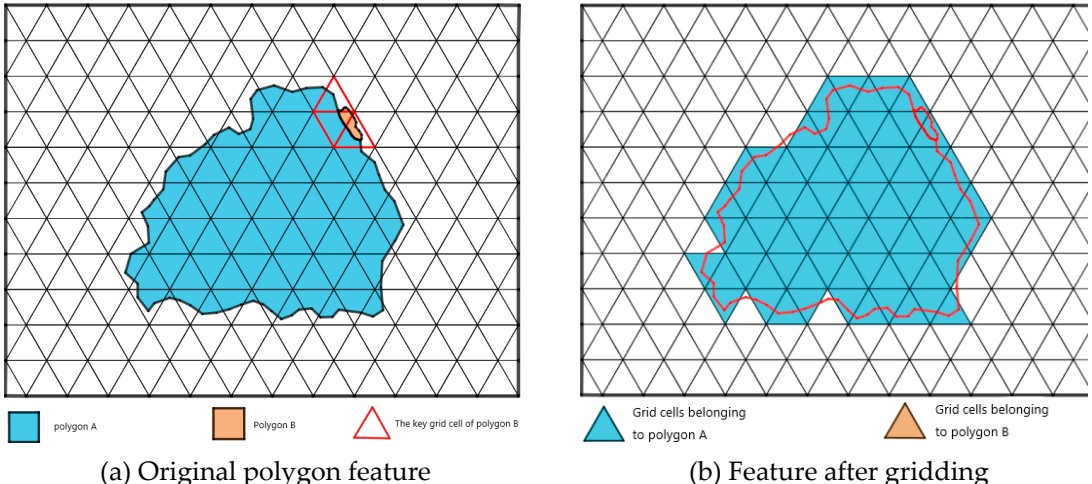

(a) Original polygon feature

(b) Feature after gridding

**Figure 7.** Polygon disappearance after gridding of polygon features.

## 5. Topology Preserving Gridding Method for Vector Features

### 5.1. Gridding of Point Features

The gridding of point feature is usually represented by a single grid cell of a specific level, and the grid cell identified by grid cell coding corresponds to the original point feature.

For the topological distortion of point features, it is only necessary to traverse all the features and compare the topological relations with other features before and after gridding. If the coordinates of the original point features i and j satisfy $x_i \neq x_j \| y_i \neq y_j$, and the grid cell encoding is $code_i = code_j$ after gridding, it is shown that the topological distortion of points i and j occurs after the initial gridding. Their topological relationship changes from disjoint to equals.

Because of the multi-resolution characteristic of DGGs, when topology distortion is repaired, the position information of the point features with topology distortion can be described by using the grid cells of a higher level until all the topological distortion is eliminated, as shown in Figure 8.

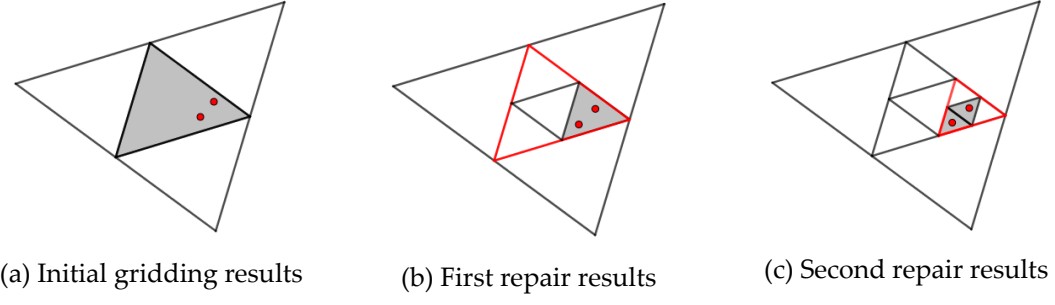

(a) Initial gridding results

(b) First repair results

(c) Second repair results

**Figure 8.** Repair of topological distortion of point features.

In Figure 8a, the two original point features are disjoint, but after the initial gridding, they are transformed into the same grid cell, and their topological relationship changes from disjoint to equals, accordingly. When the distortion is repaired, the two original point features are the first gridding with a higher-level grid cell, but the two features are still located in the same grid cell (Figure 8b); thus, the process is repeated and a higher-level sub-grid cell is used so that the original two point features

are transformed into different grid cells (Figure 8c). The topological relationship is restored to disjoint and the repair is completed.

### 5.2. Gridding of Polyline Features

A polyline feature is a one-dimensional geometric object represented by a point feature set, which is composed of connected straight line segments, in which each pair of continuous points defines a straight-line segment [19]. In DGGs, line features are usually represented by a set of continuous adjacent grid cells. Therefore, the gridding of line features is the transformation from connected geometric straight-line segments to continuous grid cell strings. In this regard, a series of mature algorithms [20] have been proposed by scholars.

After the initial gridding of line features, the topological relationship may change from disjoint to overlaps. When detecting topological distortion, it is only necessary to compare the topological relationship of the line features before and after the initial gridding. If the original polyline feature i is disjoint from j, and the intersection between the grid cell coding sets after the initial gridding is not empty, that is to say, $codeSet_i \cap codeSet_j \neq \varnothing$, the polyline feature i and j generate topological distortion after the initial gridding, which changes the relationship from disjoint to overlaps.

When repairing topological distortion, similarly, to point features, all line features with topological distortion are expressed by higher-level grid cells until all topological distortions are repaired, as shown in Figure 9.

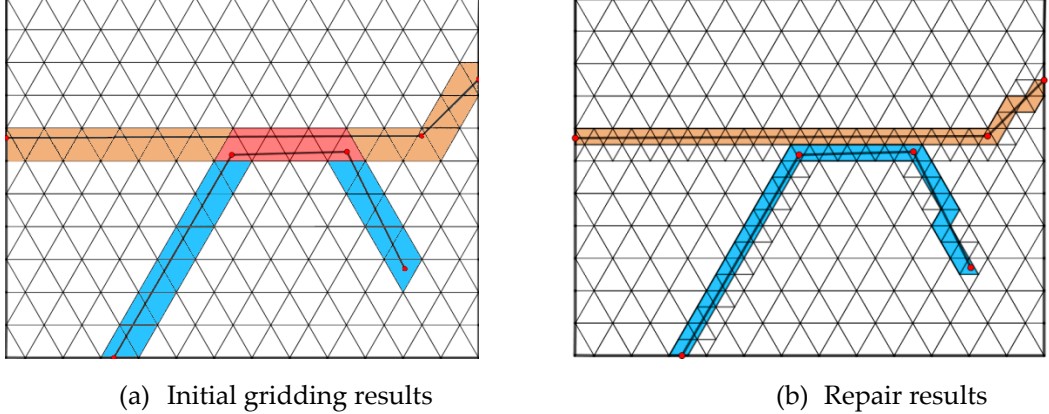

(a) Initial gridding results        (b) Repair results

**Figure 9.** Repair of topological distortion of line features.

In Figure 9a, the two original line features are disjointed, but after the initial gridding, the topological relationship changes to overlaps. When the distortion is repaired, using higher-level grid cells, it can be seen that there is no overlapping (Figure 9b) after the gridding of the two polyline features, and the topological relationship is restored from overlaps to disjoint.

### 5.3. Gridding of Polygon Features

Polygon features are two-dimensional geometric objects represented by an external boundary and zero or more internal boundaries, each of which replaces a hole in the polygon feature [19]. In DGGs, polygon features are usually expressed by a series of grid cells at the corresponding level. Therefore, in DGGs, the essence of gridding of polygon features is to use grid cells to express the region between the outer ring and the inner ring of the original polygon features.

After the initial gridding of polygon features, three topological distortions may occur, i.e., meet to disjoint, disjoint to meet, and polygon disappearance. These topological distortions usually occur in narrow areas of polygon features [21–23]. The division of grid cells in narrow areas directly determines whether the topological relationship between polygon features can be maintained correctly. In order to

identify these grid cells, the grid cells are grouped into three categories according to the positional relationship between the polygon features and the grid cells, as shown in Figure 10.

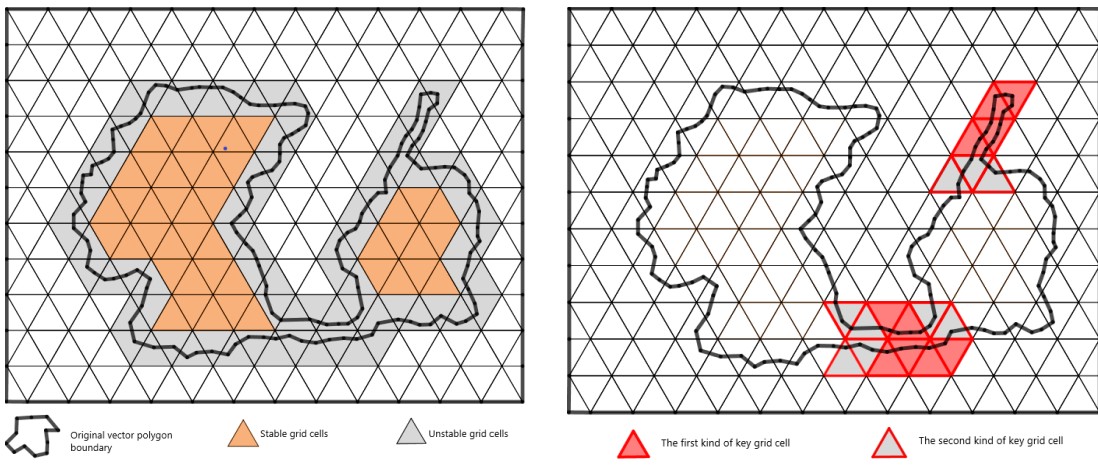

(a) Stable grid and unstable grid cells     (b) Two different types of key grid cells

**Figure 10.** Grid cell classification.

(1) Stable grid cell

$$R_9(Ploygon,\ StableCell) = \begin{bmatrix} T & F & F \\ T & * & F \\ T & F & T \end{bmatrix} \tag{7}$$

For a certain polygon feature, a grid cell that is completely contained in the range between its inner and outer rings is a stable grid cell, which is represented by the 9-intersection model as shown in Equation (7). Because the stable grid cells only intersect with a single polygon feature, if the stable grid cells of a certain polygon feature can only be attributed to that polygon according to the principle of area dominance, there is no other possibility.

(2) Unstable grid cell

$$R_9(Ploygon, VariableCell) = \begin{bmatrix} T & T & T \\ T & * & T \\ * & * & T \end{bmatrix} \tag{8}$$

For a polygon feature, a grid cell intersecting its outer ring or inner ring is an unstable grid cell, which is represented by the 9-intersection model as shown in Equation (8). Because unstable grid cells may intersect with multiple polygon features, according to the principle of area dominance, the unstable grid cells of a certain polygon feature may be attributed to that polygon or to other polygons intersecting with it.

(3) Key grid cell

For a polygon feature, a narrow area is an area with a width of less than a grid edge length in the original polygon feature. A grid cell that intersects a narrow area is a key grid cell. Further, a key grid cell that also intersects with the boundary of the polygon feature is a special kind of unstable grid cell. Because it lies in the area of unstable grid concentration, the key grid cell of a polygon feature is easily incorrectly divided, which may lead to a change of the topological relationship of the polygon.

In order to facilitate extraction, the key grid cells are divided into two types as shown in Figure 10b. If a grid cell is not a stable grid cell and the adjacent grid cell is not stable, the grid cell is the first type of key grid cell. If a grid cell is an unstable grid cell and the adjacent grid cell is the first type of key grid cell, the grid cell is the second type of key grid cell (Figure 10b).

In this paper, the most widely used principle of area dominance [24] is adopted in the initial gridding of polygon features [17,25]. In the process of initial gridding, it is necessary to record the topological adjacency relationship between the original polygon features and the stable grid cells, unstable grid cells, and key grid cells of each polygon object, so as to facilitate the subsequent topological distortion detection and repair. The specific processes are as follows (Figure 11):

**Step 1:** Input the original polygon feature layer to obtain the layer range and geometric information for each polygon feature.

**Step 2:** Create a virtual polygon. In order to preserve the topological relationship between each polygon feature and blank areas with no data and at the same time to provide a consistent processing process for all polygons, a virtual polygon is created to represent a blank area that is not covered by actual polygon features in the layer. A rectangle is created according to the layer range, and then, all the original actual polygon features are erased (difference) sequentially so that the virtual polygon representing the blank area is obtained.

**Step 3:** Create the adjacency matrix $M_o$ of $n \times n$, where $n$ is the number of polygon features. The topological adjacency relationships between the original polygon features are calculated and assigned to $M_o$. If the polygon $i$ is adjacent to the polygon $j$, $M_o[i][j] = M_o[j][i] = 1$, otherwise $M_o[i][j] = M_o[j][i] = 0$.

**Step 4:** Traverse the various polygon features. For all grid cells in the range of the feature's bounding box, the type of grid cell is judged according to the position relationship between the grid cell and polygon feature. If a grid cell is completely included in a polygon, it is classified as a stable grid cell of that polygon, and if it intersects the boundary of the polygon, the grid cell is classified as an unstable grid cell of the polygon.

**Step 5:** Obtain the key grid cells of each polygon. According to the characteristics of the key grid cells, the first type of key grid cell is first screened from the unstable grid cells, and then, the second type of key grid cell is selected from the unstable grid cells.

**Step 6:** Determine the initial ownership of all grid cells. For stable grid cells, because the owner is unique, the stable grid cells on a certain polygon are assigned to that polygon; for unstable grid cells (including key grid cells), the overlapping area with all the intersecting polygons is calculated, and according to the principle of area dominance, it is classified as the polygon with the largest overlapping area.

**Step 7:** At this point, all the grid cells' ownerships have been identified. All grid cells and their ownerships are output, and the initial gridding of the polygon feature ends.

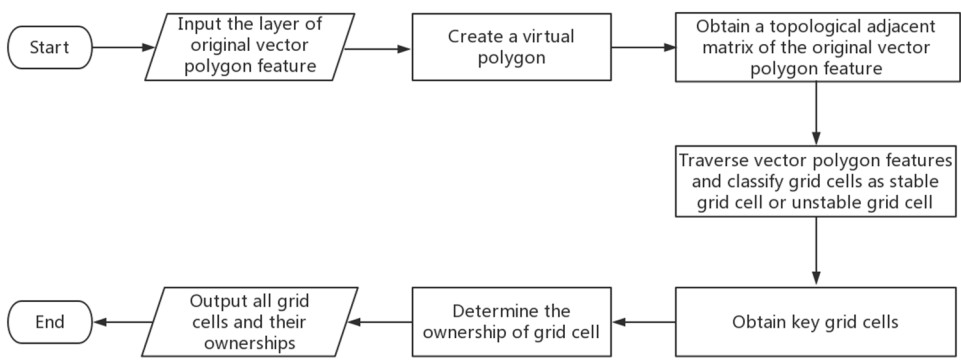

**Figure 11.** Initial gridding of polygon features.

After the initial gridding, the original polygon features have been transformed into grid cell set representation, but the topological relationships between polygons may have changed. Topological distortion needs to be detected according to two adjacency matrices, $M_o$ and $M_g$.

The matrix $M_o$, which records the topological adjacency of the original polygon features, is obtained in the initial gridding process. The matrix $M_g$ records the topological adjacency relation between the

polygons after the initial gridding. It is a calculator from the unstable grid that is recorded in the initial gridding process. In the case of an unstable grid, it is assumed that the unstable grid cells are assigned to the polygon $i$, and an adjacent grid cell is assigned to the polygon $j(i \neq j)$; it is known that the polygon $i$ after gridding is adjacent to the polygon $j$, and thus, $M_g[i][j] = M_g[j][i] = 1$. After traversing all the unstable grid cells, the topological relation adjacent matrix $M_g$ between all polygons can be obtained after gridding.

Three kinds of topological distortions after gridding can be detected by using the adjacency matrices $M_o$ and $M_g$. If $M_o[i][j] = 1$ and $M_g[i][j] = 0$, then the topological relationship between polygon $i$ and polygon j changed from meet to disjoint, and if $M_o[i][j] = 0$ and $M_g[i][j] = 1$, the topological relationship between polygon i and polygon j changed from disjoint to meet. If for polygon i, it is not adjacent to any polygon after gridding, that is, $\forall M_g[i][j] = 0(0 \leq j < n)$, then the polygon i disappears after gridding.

The misclassification of key grid cells leads to the above topological distortion, and the fundamental reason why key grid cells are incorrectly divided is that the expression accuracy of the narrow region of the polygon feature is not of high enough resolution. Thus, the above topological distortion can be repaired by using a higher-level grid cell to express the polygon. First, the key grid cells of the polygon feature with topological distortion are replaced with high-level grid cells. After that, the ownership of all new grid cells is determined according to the principle of area dominance, and the key grid cells are re-determined. Finally, the topological distortion detection is performed again until all topological distortions are repaired, as shown in Figure 12.

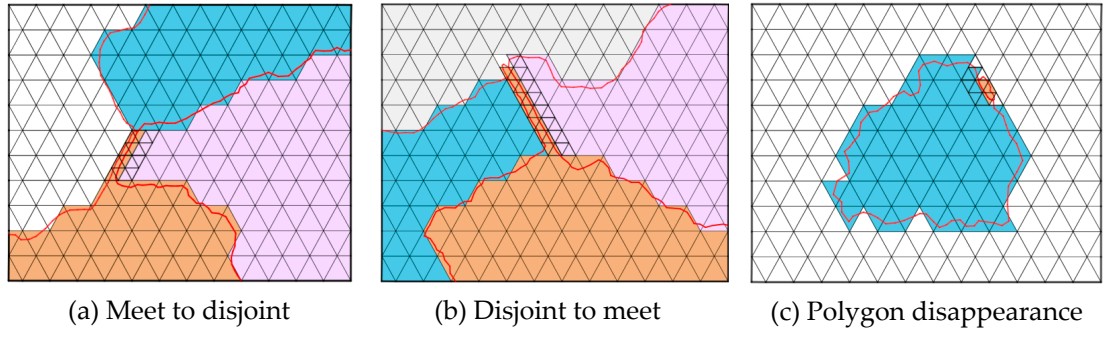

(a) Meet to disjoint        (b) Disjoint to meet        (c) Polygon disappearance

**Figure 12.** Four topological distortion repair examples.

## 6. Case Study

In order to verify the topology-preserving gridding method for vector features, an icosahedral triangular grid based on the SQT partition model [26] is used to perform the experiment. The map scale corresponding to each level of the grid is shown in Table 1. In terms of experimental data, some vector data with a scale of 1:110,000,000 downloaded using Natural Earth (https://www.naturalearthdata.com/) are selected. From Table 1, the scale of 1:110,000,000 corresponds to the subdivision level 9, and we also carry out experiments on the adjacent level 8 and 10 grids. Original vector features and gridding results for subdivision level 8 are shown in Figure 13.

**Table 1.** Map scale for each discrete level of the regular icosahedral triangular grid.

| Grid Level | Average Grid Edge Length (/m) | Map Scale SD |
|:---:|:---:|:---:|
| 7 | 55,106.47 | 1:550,000,000 |
| 8 | 27,553.24 | 1:300,000,000 |
| 9 | 13,776.62 | 1:100,000,000 |
| 10 | 6888.31 | 1:50,000,000 |
| 11 | 3444.16 | 1:35,000,000 |

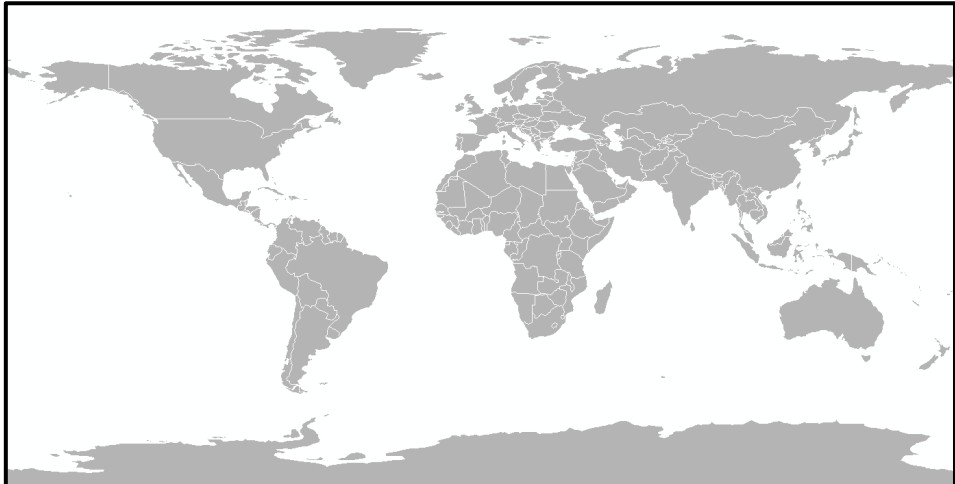

(**a**) Primitive polygon feature data.

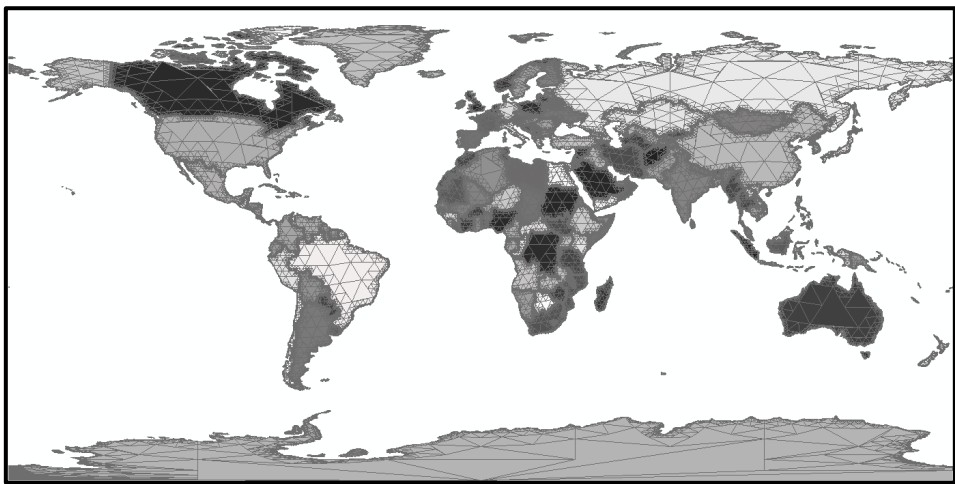

(**b**) The result of gridding polygon features.

**Figure 13.** Polygon feature data and gridding result.

The results of the experiment are evaluated from the two aspects of the topological distortion repair rate and the result data volume. Additionally, the results are compared with the method of increasing the resolution of the whole grid. To determine the topological distortion repair rate, the topology-preserving gridding method is used for vector point, line, and polygon features, respectively; the count of topological distortions before and after the repair is counted, and the repair rate is calculated. For the determination of the data volume, the method of increasing the global grid resolution and the proposed method are used to mesh the same experimental data, and the data volume of different levels is compared.

*6.1. Experimental Study on Repair Rate of Topological Distortion*

Using the topology-preserving gridding method, the selected experimental data is gridded, and the number of topological distortions before and after topological distortion repair are counted. The results are as shown in Table 2. Figures 14–16 are examples of topological distortion repair at level 8.

**Table 2.** Number of topological distortions before and after repair and repair rate.

| Level | Feature Type | Before Repair | After Repair | Rate Repair |
|---|---|---|---|---|
|  | Point | 128 | 0 | 100 |
| 8 | Polyline | 14 | 0 | 100 |
|  | Polygon | 71 | 0 | 100 |
|  | Point | 28 | 0 | 100 |
| 9 | Polyline | 4 | 0 | 100 |
|  | Polygon | 31 | 0 | 100 |
|  | Point | 8 | 0 | 100 |
| 10 | Polyline | 3 | 0 | 100 |
|  | Polygon | 12 | 0 | 100 |

From the experimental results of topological distortion repair and the repair examples, it can be seen that the topological distortion after gridding can be effectively repaired by the topology-preserving gridding method for vector features. Through the multi-level resolution characteristic of DGGs, topological distortion can be repaired recursively by increasing the resolution of the object whose topological relationship has changed; thus, the change of topological relationship after entity object gridding can be completely avoided.

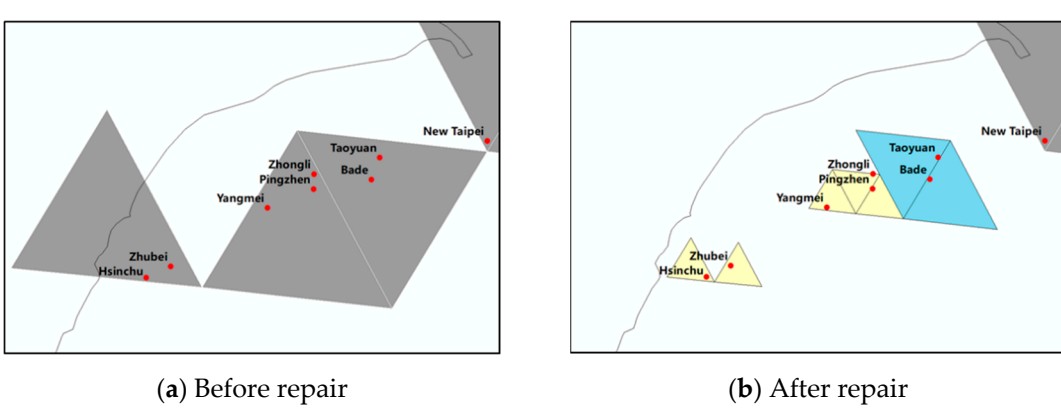

(**a**) Before repair

(**b**) After repair

**Figure 14.** An example of topology distortion repair of point features by gridding.

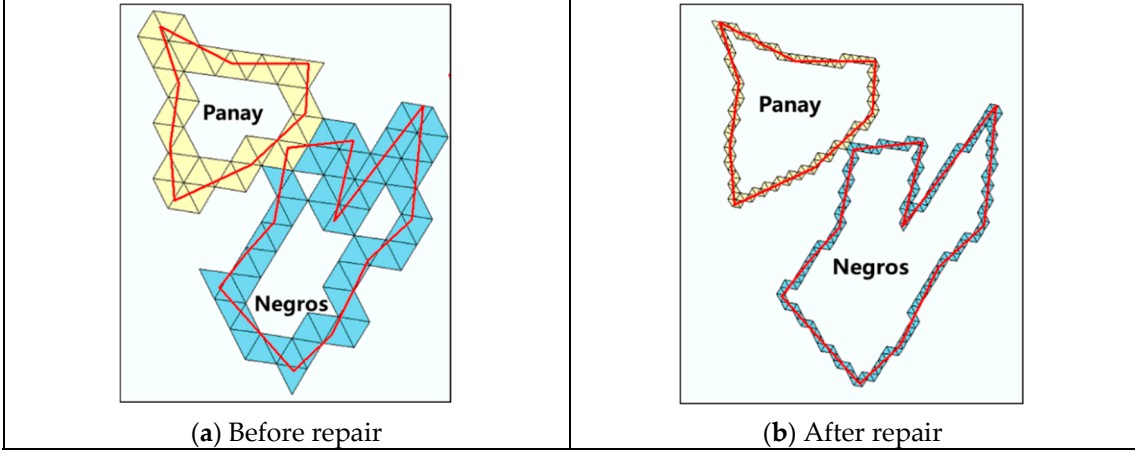

(**a**) Before repair

(**b**) After repair

**Figure 15.** An example of topology distortion repair of line features by gridding.

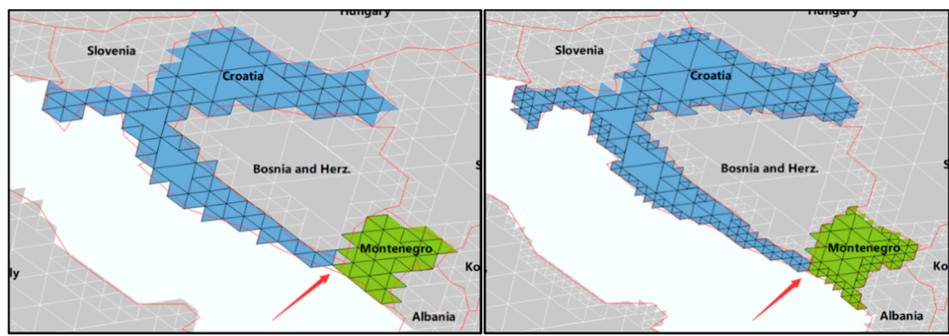

(**a**) Before and (**b**) after meet to disjoint repair.

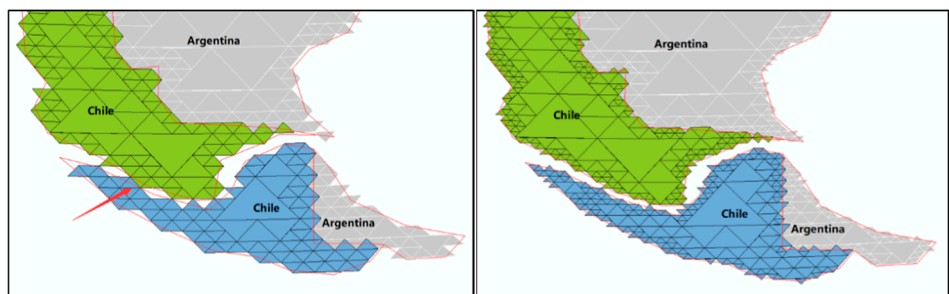

(**c**) Before and (**d**) after disjoint to meet repair.

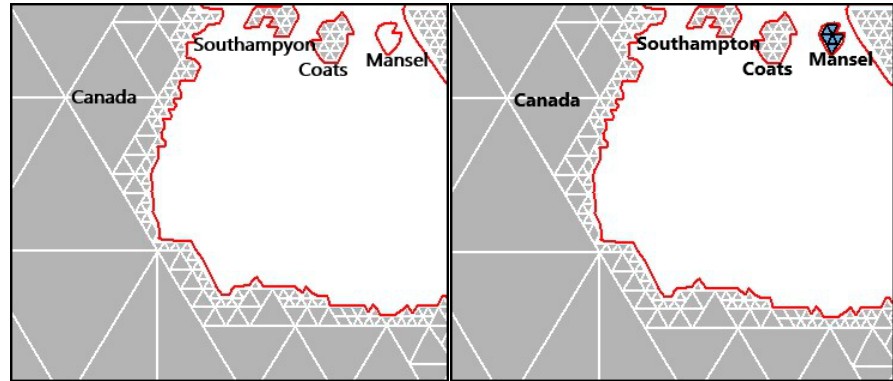

(**e**) Before and (**f**) after polygon disappearance repair.

**Figure 16.** Three examples of topology distortion repair.

*6.2. Data Volume Experiment*

The method of avoiding topological distortion by increasing the resolution of the whole grid is used to gridding the selected experimental data, and then, the data volume of the whole grid method is compared with the method in this paper, as shown in Table 3 and Figure 17.

**Table 3.** Data volume of the gridding results of the three methods (kB).

| Repair Method | Level 8 | | | Level 9 | | | Level 10 | | |
|---|---|---|---|---|---|---|---|---|---|
| | Point | Line | Polygon | Point | Line | Polygon | Point | Line | Polygon |
| None | 1333 | 138 | 572 | 1348 | 219 | 1199 | 1369 | 380 | 2480 |
| Global | 1412 | 2625 | 5061 | 1412 | 2625 | 5061 | 1412 | 2625 | 5061 |
| Proposed | 1334 | 213 | 574 | 1348 | 220 | 1200 | 1369 | 381 | 2481 |

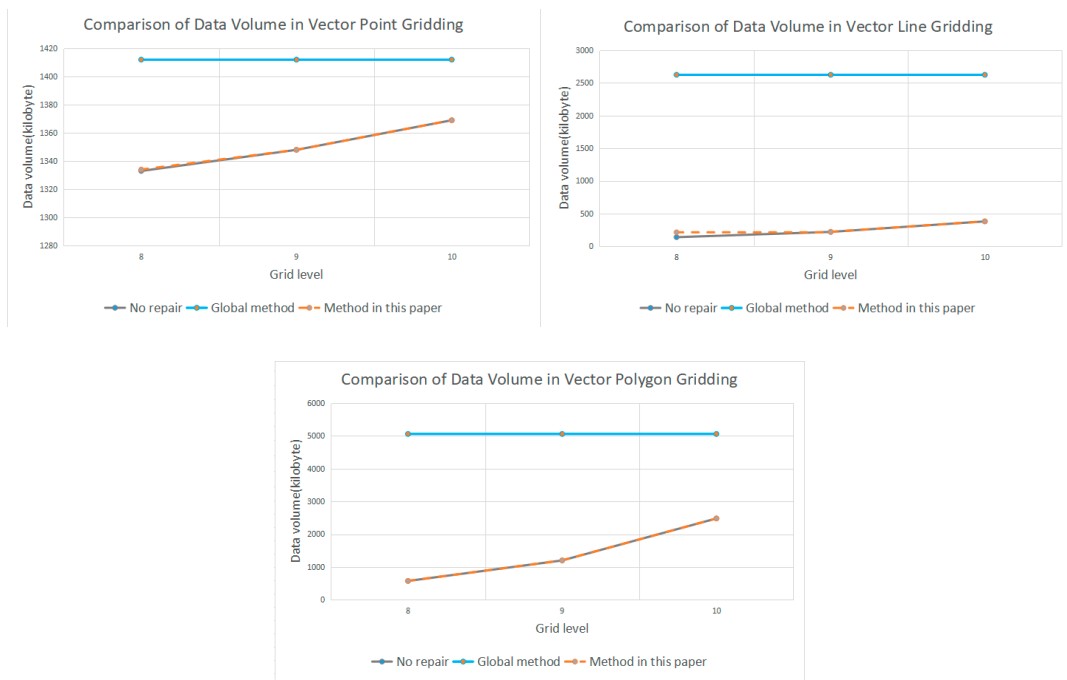

**Figure 17.** Comparison of the data volumes of the gridding results of the three methods.

From the experimental results for data volume, it can be seen that compared with the method of increasing global grid resolution, the data volume added by the method in this paper is negligible. The topology-preserving gridding method for vector features determines objects whose topological relationships change by topological distortion detection. According to the multi-level resolution characteristics, only the resolution of these objects is increased recursively. Therefore, compared with the global grid resolution method, the data volume is greatly reduced while maintaining the topological relationship of the original vector features.

*6.3. Algorithm Efficiency Experiment*

Using the same data as in data volume expreriment, the execution time of the whole grid method is compared with the method in this paper, as shown in Table 4 and Figure 18.

**Table 4.** Execution time of the gridding results of the three methods (second).

| Repair Method | Level 8 | | | Level 9 | | | Level 10 | | |
|---|---|---|---|---|---|---|---|---|---|
| | Point | Line | Polygon | Point | Line | Polygon | Point | Line | Polygon |
| None | 0.3 | 0.3 | 115.6 | 0.4 | 0.4 | 451.1 | 0.4 | 0.4 | 1780.0 |
| Global | 5.1 | 14.6 | 7133.8 | 5.1 | 14.6 | 7133.8 | 5.1 | 14.6 | 7133.8 |
| Proposed | 4.8 | 0.9 | 119.5 | 4.9 | 1.0 | 461.3 | 5.0 | 1.7 | 1836.4 |

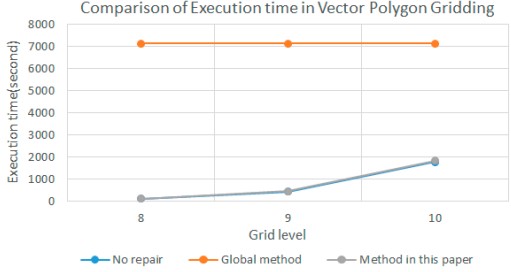

**Figure 18.** Comparison of the execution time of the gridding results of the three methods.

From the experimental results for execution time, although the topology-preserving gridding method in this paper slightly increases the execution time but only increases the grid resolution locally, the algorithm efficiency is much higher than the global method.

## 7. Discussion

### 7.1. Comparison with the Global Method

Similar to the traditional gridding method, the proposed method first determines the initial gridding level by the map scale of the original vector features and then uses different initial gridding strategies for different types of vector features. This paper describes a topological distortion detection method according to the grid cell relationship in the grid system and a method to repair the topological distortion by employing the multi-level resolution characteristics, which is the innovation of the topology-preserving gridding method for vector features. The proposed method of this paper detects objects whose topological relationship has changed by topological distortion detection, and it selectively increases their local resolution recursively according to the multi-level resolution characteristics. Thus, the data volume and execution time is greatly reduced compared with the method of increasing the resolution of the global grid.

However, compared with the global method, the proposed method divides the grid into several different types to deal with so as to repair the topological distortion without significantly improving the amount of data, so the algorithm is difficult to implement.

### 7.2. Extensibility

In this paper, the icosahedral triangular grid is selected to describe and verify the proposed method. However, the method is applicable to other DGGs, such as spherical tetrahedral grids, hexagonal grids, and traditional longitude-latitude grids. In order to employ this method in a spherical quadrangle grid, hexagonal grid, or longitude-latitude grid, only the adjacent relation search algorithm and level relation search algorithm, remaining algorithms, do not require modification. Therefore, the core idea of this method can be applied to other DGGs.

## 8. Conclusion

In view of the change of topological relationships of vector features during the integration of DGGs, this paper proposes a topology-preserving gridding method for vector features in DGGs. In this method, the initial gridding level is determined by the map scale of the original vector feature, and then, the corresponding initial gridding strategy is used for different types of vector features. The topological distortion is detected according to the relationship of grid cells in the grid system, and finally, the topological distortion is repaired by employing the characteristic of multi-level resolution. The experimental results show that the topology-preserving gridding method for vector features can not only effectively maintain the topological relationship between the original vector features but also minimizes data volume growth.

**Author Contributions:** Conceptualization, Wenjie Lian; funding acquisition, Yudi Zhang and Bingxian Lin; methodology, Liangchen Zhou, Wenjie Lian, and Bingxian Lin; writing—original draft, Liangchen. All authors have read and agreed to the published version of the manuscript.

**Funding:** This research was funded by National Key Research and Development Program of China: 2018YFB0505301 and National Natural Science Foundation of China: 41571381.

**Conflicts of Interest:** The authors declare no conflict of interest.

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
