# Peer review of "A Topology Preserving Gridding Method for Vector Features in Discrete Global Grid Systems"

_ijgi, doi:10.3390/ijgi9030168_

Round 1

Reviewer 1 Report

This paper presents a new idea and methodology to prevent the distortion of topological structures in global grid systems. Overall, the experiment is well designed and the results of the analysis show the usefulness of the proposed methodology. However, the topics covered in this paper are also consistent with the various operations of Map generalization in cartography. So please explain the relationship between the research topic and map generalization. It is also worth discussing the applicability of the methodology to map generalization.

The following are minor modification requirements.
-Citation notation in the text is incorrect. Please check the number of authors cited in the reference and correct the citations.
Page 2 Egenhofer's 9-intersection model is considered a key part of our methodology. A brief description of what this method is all about may help improve the understanding of the reader.

Author Response

Reviewer: 1

  1. please explain the relationship between the research topic and map generalization.

Response: Map generalization is a process of deriving maps at a smaller scale from those at a larger scale. In the process, topological relations between map features on a spatial representation may have changed after a geometric transformation. The common resolutions are proposed to change map features by some operations, such as displacement, selection and merge. So, it is different from this research topic that focus on changing the discreate grid cells to preserve the topological relation.

  1. It is also worth discussing the applicability of the methodology to map generalization.

Response: Although this research is used to modify the grid cells not vector features, it might be used to detect the topological conflicts after map generalization. However, it is not the main topic of this research, and it is not suitable to discuss it in a large paragraph.

  1. Citation notation in the text is incorrect. Please check the number of authors cited in the reference and correct the citations.

Response: citation related errors are Corrected.

  1. Page 2 Egenhofer's 9-intersection model is considered a key part of our methodology. A brief description of what this method is all about may help improve the understanding of the reader.

Response: A description of 9-intersection model has been added in the section 3.

Reviewer 2 Report

This paper proposes a topology preserving gridding method for vector features in DGGs, having in mind the change of topological relationships of vector features during integration of DGGs. The authors presents a method in which the initial gridding level is determined by the map scale of the original vector feature, and then the corresponding initial gridding strategy is used for different types of vector features.
The topological distortion is detected according to the relationship of grid cells in the grid system, and finally the topological distortion is repaired by employing the characteristic of multi-level resolution. The experimental results show that the topology preserving gridding method for vector features can not only effectively maintain the topological relationship between the original vector features, but also minimizes data volume growth.
Overview of related research about DGG / multi-resolution global grids would increase readability of paper.
Table 2. Label in row after "Rate", like something is missing, but it is unusual that table presents same values in 2 rows, sow same information can be explained in different way.
What would be dra
wbacks of the approach? Is there any?
Evaluation in terms on numerical precission would help (increase of precision)?
Additional use cases can be mentiond, where this approach would be helpfull.

Author Response

Reviewer: 2

  1. Overview of related research about DGG / multi-resolution global grids would increase readability of paper.

Response: An overview of recent research about DGGs has been added in the section 2.

  1. Table 2. Label in row after "Rate", like something is missing, but it is unusual that table presents same values in 2 rows, sow same information can be explained in different way.

Response: Table 2 header error and table structure modified.

  1. What would be drawbacks of the approach? Is there any?

Response: A description about drawbacks of the approach has been added at the bottom of section 7 (1).

  1. Evaluation in terms on numerical precision would help (increase of precision)?

Response: This paper focuses on the accuracy of topology rather than numerical precision.

  1. Additional use cases can be mentioned, where this approach would be helpful.

Response: A description of an application in ocean model has been added in section 1.
